# Histone demethylase LSD1 promotes RIG-I poly-ubiquitination and anti-viral gene expression

Qi-Xin Hu[1,2], Hui-Yi Wang[1,2], Lu Jiang[1,2], Chen-Yu Wang[1,2], Lin-Gao Ju[3], Yuan Zhu[3], Bo Zhong[1,4], Min Wu[1,2]*, Zhen Wang[1,2]*, Lian-Yun Li[1,2]*

**1** Frontier Science Center for Immunology and Metabolism, College of Life Sciences, Wuhan University, Wuhan, Hubei, China, **2** Hubei Key Laboratory of Cell Homeostasis, Hubei Key Laboratory of Developmentally Originated Disease, Hubei Key Laboratory of Enteropathy, Wuhan University, Wuhan, Hubei, China, **3** Department of Biological Repositories, Zhongnan Hospital of Wuhan University, Wuhan, Hubei, China, **4** Department of Immunology, Medical Research Institute, School of Medicine, Wuhan University, Wuhan, China

* wumin@whu.edu.cn (MW); wildman1992@whu.edu.cn (ZW); lilianyun@whu.edu.cn (L-YL)

**Data Availability Statement:** All relevant data are within the manuscript and its Supporting Information files.

## Abstract

Under RNA virus infection, retinoic acid-inducible gene I (RIG-I) in host cells recognizes viral RNA and activates the expression of type I IFN. To investigate the roles of protein methyltransferases and demethylases in RIG-I antiviral signaling pathway, we screened all the known related enzymes with a siRNA library and identified LSD1 as a positive regulator for RIG-I signaling. Exogenous expression of LSD1 enhances RIG-I signaling activated by virus stimulation, whereas its deficiency restricts it. LSD1 interacts with RIG-I, promotes its K63-linked polyubiquitination and interaction with VISA/MAVS. Interestingly, LSD1 exerts its function in antiviral response not dependent on its demethylase activity but through enhancing the interaction between RIG-I with E3 ligases, especially TRIM25. Furthermore, we provide *in vivo* evidence that LSD1 increases antiviral gene expression and inhibits viral replication. Taken together, our findings demonstrate that LSD1 is a positive regulator of signaling pathway triggered by RNA-virus through mediating RIG-I polyubiquitination.

## Author summary

RIG-I signaling pathway is critical for human cells to defend from RNA virus infection, such as SARS-CoV-2, influenza virus, and Vesicular Stomatitis Virus (VSV). LSD1 is a histone demethylase regulating transcription. The current study reveals a novel function of LSD1 in regulating the activation of RIG-I signaling pathway. LSD1 interacts with RIG-I and promotes RIG-I poly-ubiquitination independent of its demethylase activity. LSD1 facilitates the interaction between RIG-I and its ubiquitin E3 ligase TRIM25, which is crucial for recruitment of downstream proteins. The mice with LSD1 deficiency are susceptible to virus infection and have lower survival rate. Taken together, our findings demonstrate a novel molecular mechanism for regulating the anti-viral RIG-I signaling pathway.

**Funding:** This work was supported by the Ministry of Science and Technology of China (2016YFA0502100) (MW), the Fundamental Research Funds for the Central Universities (MW), National Natural Science Foundation of China to MW (31771503 and 81972647) and LL (31670874), Science and Technology Department of Hubei Province of China (2017ACA095) (MW). The funders had no role in study design, data collection and analysis, decision to publish, or preparation of the manuscript.

**Competing interests:** The authors have declared that no competing interests exist.

## Introduction

Innate immunity is the first barrier against pathogen invasion, such as viruses and bacteria [1–3]. Pathogens are recognized by specific pattern recognition receptors, which induce a cascade of immune responses, and finally eliminated by interferons and cytokines [4,5]. Several families of pattern recognition receptors have been identified, including Toll-like receptors (TLRs), retinoic acid–inducible gene I (RIG-I)–like receptors (RLRs), cyclic GMP–AMP synthase (cGAS) like receptors, NOD-like receptors, and C-type lectin receptors [2,3]. RNA viruses, such as Sendai virus (SeV) and vesicular stomatitis virus (VSV), are recognized by RLRs when infecting cells. RIG-I, melanoma differentiation-associated gen 5 (MDA5) and DExH-box helicase 58 (DHX58, also known as LGP2) are three types of RLRs, and they all belong to the DEA(D/H) box RNA helicase family [6,7]. RIG-I recognizes the 5' end triphosphate of RNA virus genome or double-stranded RNA to sense virus infection [8–10]. After that, RIG-I interacts with virus-induced signaling adaptor (VISA, also called MAVS or IPS1) to stimulate it. Activated VISA/MAVS promotes phosphorylation of TANK binding kinase 1 (TBK1) and the subsequent phosphorylation of interferon regulatory factor 3 (IRF3). Phosphorylated IRF3 enters nucleus to turn on the expression of type I interferons [4,6,7,11–18].

Factors of innate immune system are often modified post-translationally, such as polyubiquitination and phosphorylation, which is important for pathway activation [19–22]. For example, K63-linked polyubiquitination on RIG-I is required for the activation of downstream signaling pathway [23–25]; On the other hand, RIG-I with K48-linked polyubiquitination is degraded to restrict immune response at late period of antiviral response [26,27]. RGI-I's ubiquitination is fine-tuned by ubiquitin ligases and deubiquitinating enzymes (DUBs) [28,29]. Tripartite motif containing 4 (TRIM4), tripartite motif containing 25 (TRIM25) and ring finger protein 135 (RNF135) are responsible for K63-linked polyubiquitination of RIG-I, whereas CYLD lysine 63 deubiquitinase (CYLD) negatively regulates it through deubiquitination [25,30–34]. Ring finger protein 125 (RNF125) is essential for RIG-I K48-linked ubiquitination and the subsequent degradation [27]. In the current study, we report lysine demethylase 1A (KDM1A, also called LSD1), a demethylase for histone H3, promotes activation of RIG-I signaling pathway through enhancing RIG-I's K63-linked polyubiquitination.

LSD1 is the first reported 'eraser' of histone methylation which demethylates lysine 4 (H3K4) and lysine 9 (H3K9) on histone H3 [35–38]. Besides, LSD1 has a wide range of functions in many cell processes, such as autophagy, differentiation, tumorigenesis, as well as immunity [39–49]. Interestingly, LSD1 exhibits opposite functions during infections of different viruses. As reported, inhibition of LSD1 restricts herpesvirus infection, shedding, and recurrence, through epigenetic suppression of viral genomes [43], while LSD1 represses influenza A virus infection by erasing monomethylation on lysine 88 of interferon induced transmembrane protein 3 (IFITM3) [40]. IFITM3 is an IFN-induced antiviral protein which inhibits the entry of viruses to host cells by preventing viral fusion with cholesterol depleted endosomes [40,50–52]. In the current study, we found that LSD1 serves as a positive regulator of RIG-I signaling pathway activated by RNA virus, which further expands the important roles of LSD1 in anti-viral innate immunity.

## Materials and methods

### Ethics statement

Mice were maintained in the special pathogen-free facility of College of Life Sciences at Wuhan University. All the animal operations were following the laboratory animal guidelines of Wuhan University and approved by the Animal Experiments Ethics Committee of

Wuhan University (Protocol NO. 14110B). No patient study was involved and the consent to participate is not applicable.

## Cell lines, tissue culture and transfection

HEK293T cells, L929 cells and Vero cells were purchased from the Cell Bank of Chinese Academy and were cultured in DMEM (Gibco) supplemented with 10% FBS (Biological Industries) and 1% penicillin and streptomycin (HyClone). HEK293T cells were transfected with Lipofectamine 2000 (Invitrogen) or calcium phosphate. SeV and VSV-GFP were gifted by Dr. Hong-Bing Shu of Wuhan University, and then amplified in our lab.

Transfection of 293T was performed with a standard calcium phosphate precipitation method [53]. Transfection of siRNAs and other cell lines was performed with lipofectamine 2000 (Invitrogen) with the manufacturer's protocol.

## Mice

*Lsd1* $^{f/+}$ mice on the C57BL/6 background were purchased from Jackson lab. *Lyz2*-Cre mice C57BL/6J were gifted from Dr. Bo Zhong (College of Life Sciences, Wuhan University,). Mice genotype identification was performed by polymerase chain reaction (PCR) analysis of DNA isolated from the tail using the following primers: Kdm1a-loxp-F, GCTGGATTGAGTTGGTT GTG; Kdm1a-loxp-R, CTGCTCCTGAAAGACCTGCT; Lyz2-cre-F, GCCTGCATTACCGGT CGATGC; Lyz2-cre-R, CAGGGTGTTATAAGCAATCCC.

## Antibodies and reagents

Antibodies and reagents were purchased from the indicated merchants: Monoclonal antibodies against β-actin (Abclonal, AC026, RRID: AB_2768234), RIG-I (Abclonal, A0550, RRID: AB_2757259), LSD1 (Abclonal, A1156, RRID: AB_2721240), TBK1 (CST, 3504, RRID: AB_2255663), TBK1 pS172 (CST, 5483, RRID: AB_10693472), IRF3 (CST, 11904, RRID: AB_2722521; Santa Cruz, sc-9082, RRID: AB_2264929), IRF3 pS396 (CST, 4947, RRID: AB_823547), Myc-tag (Abclonal, AE010, RRID: AB_2770408), GFP (Abclonal, AE012, RRID: AB_2770402), HA-tag (Origene, TA180128, RRID: AB_2622290), Flag-tag (Sigma-Aldrich, F3165, RRID: AB_259529), Low Molecular Weight Poly (I:C) (Invivogen, Cat# tlrl-picw).

## RNA interference, reverse transcription and quantitative RT-PCR

The indicated cells were transfected with small interfering RNA (siRNA, LSD1 siRNA 1#: AAGGAAAGCUAGAAGAAAAUU, 2#: CAGAAGGCCUAGACAUUAAUU) and were scraped down and collected by centrifugation. Total RNA was extracted with an RNA extraction kit (Aidlab or CWBIO) according to the manufacturer's instructions. Approximately 1 ug total RNA was used for reverse transcription with a first-strand cDNA synthesis kit (Vazyme). The amount of mRNA was assayed by quantitative PCR. Data shown are the relative abundance of the indicated mRNAs normalized to that of ACTB mRNA (*Homo sapiens*) or *Gapdh* mRNA (*Mus musculus*). The sequences of primers are shown in S1 Table. Assays were repeated at least three times.

## Co-immunoprecipitation and ubiquitination analysis

HEK293T cells were harvested and lysed in NP-40 lysis buffer (50 mM Tris-HCl [pH 7.4], 150 mM NaCl, 0.5% NP-40) with proteinase inhibitors. For ubiquitination assays, cells were lysed in RIPA lysis buffer (50 mM Tris-HCl [pH 7.4], 150 mM NaCl, 0.1% SDS, 0.5% NP40, 0.5% sodium deoxycholate) with proteinase inhibitors and then sonicated for 1 minute. The

supernatant was then incubated with protein G beads (GE Healthcare) and desired antibodies at 4˚C for 4 h. The beads were spun down and washed three times with lysis buffer. The final drop of wash buffer was vacuumed out and SDS loading buffer was added to the beads, followed by immunoblotting.

## Immunoblotting analysis

Protein extracts were separated by SDS-PAGE and transferred to nitrocellulose filter membranes. After blocking with 5% skim milk in Tris-buffered saline and 0.1% Tween (TBS-T), the membranes were incubated with the desired primary antibodies overnight at 4˚C. Then the membranes were washed and incubated with the appropriate secondary antibodies at room temperature for 1 h. The blots were detected by Clarity Western ECL Substrate (BIO-RAD). For cell assay, at least triplicate biological experiments were performed.

## Preparations of BMDMs, BMDCs and MLFs

For preparation of BMDMs, bone marrow cells were cultured in 10% M-CSF-containing conditional medium from L929 cells for 3–5 days. For preparation of BMDCs, bone marrow cells were cultured in medium containing murine GM-CSF (50 ng/mL) for 6–8 days. Primary lung fibroblasts were isolated from lungs of 8-week old mice. In brief, lungs were minced and digested in calcium and magnesium free HBSS containing 10 mg/mL type II collagenase and 20 mg/mL DNase I for 1 hour at 37˚C. Cell suspension was centrifuged at 1500 rpm for 5 min. The cells were then plated in DMEM containing 10% FBS, 1% penicillin and streptomycin.

## Plaque forming unit (PFU) assay

Vero cells were seeded into 12-well plates 1 day before measurement. Supernatants of VSV-infected cells were frozen and thawed twice, then centrifuged at 12,000 rpm for 5 min. Supernatants were serially diluted on the monolayer of Vero cells for 2 days, and PFU was measured.

## H&E staining

Tissues were isolated from indicated male mice, rinsed in ice-cold PBS, fixed in 4% poly-form-aldehyde for 48 h at 4˚C, then dehydrated, embedded in paraffin and sectioned. 4 μm tissue sections were used for staining. The tissue sections were deparaffinized and then rehydrated. After staining with Mayer's hematoxylin dye for 5–7 min, the sections were washed with water, sealed with neutral gum after air drying, and finally examined with Olympus BX51 microscope.

# Results

## LSD1 is required for activation of RIG-I signal pathway

To investigate novel mechanisms regulating RIG-I pathway, we screened a siRNA library targeting all the known methyltransferases and demethylases for genes regulating the activation of RIG-I signaling pathway. The library was designed in the lab and synthesized by the company, which contains all known and predicted protein methyltransferases and demethylases, as well as their related genes. WDR82 and TTLL12 were identified from the screen and reported as repressors of the pathway [54,55]. LSD1/KDM1A, a demethylase of histone H3K4 and H3K9, was also identified from the screen. To confirm the discovery, two independent siRNAs were used to knock down *LSD1* in HEK293T cells, and quantitative RT-PCR result indicated that the mRNA levels of *IFN-β* and two IFN-induced genes (*ISG54* and *ISG56*)

induced by SeV decreased after LSD1 depletion (Fig 1A–1C). Similarly, knockdown of *LSD1* impaired poly(I:C)-induced expression of *IFNB1*, *ISG54* and *ISG56* in HEK293T cells (Fig 1B and 1C). To further confirm the role of LSD1 in the innate immune response to RNA viruses, a HEK293T cell line with stable-expressed HA-LSD1 was constructed (Fig 1D). Consistently, exogenous expression of LSD1 enhanced SeV-induced expression of *IFNB1*, *ISG54* and *ISG56* in cells (Fig 1E). Similar results were observed in cells induced by poly(I:C) treatment (Fig 1F).

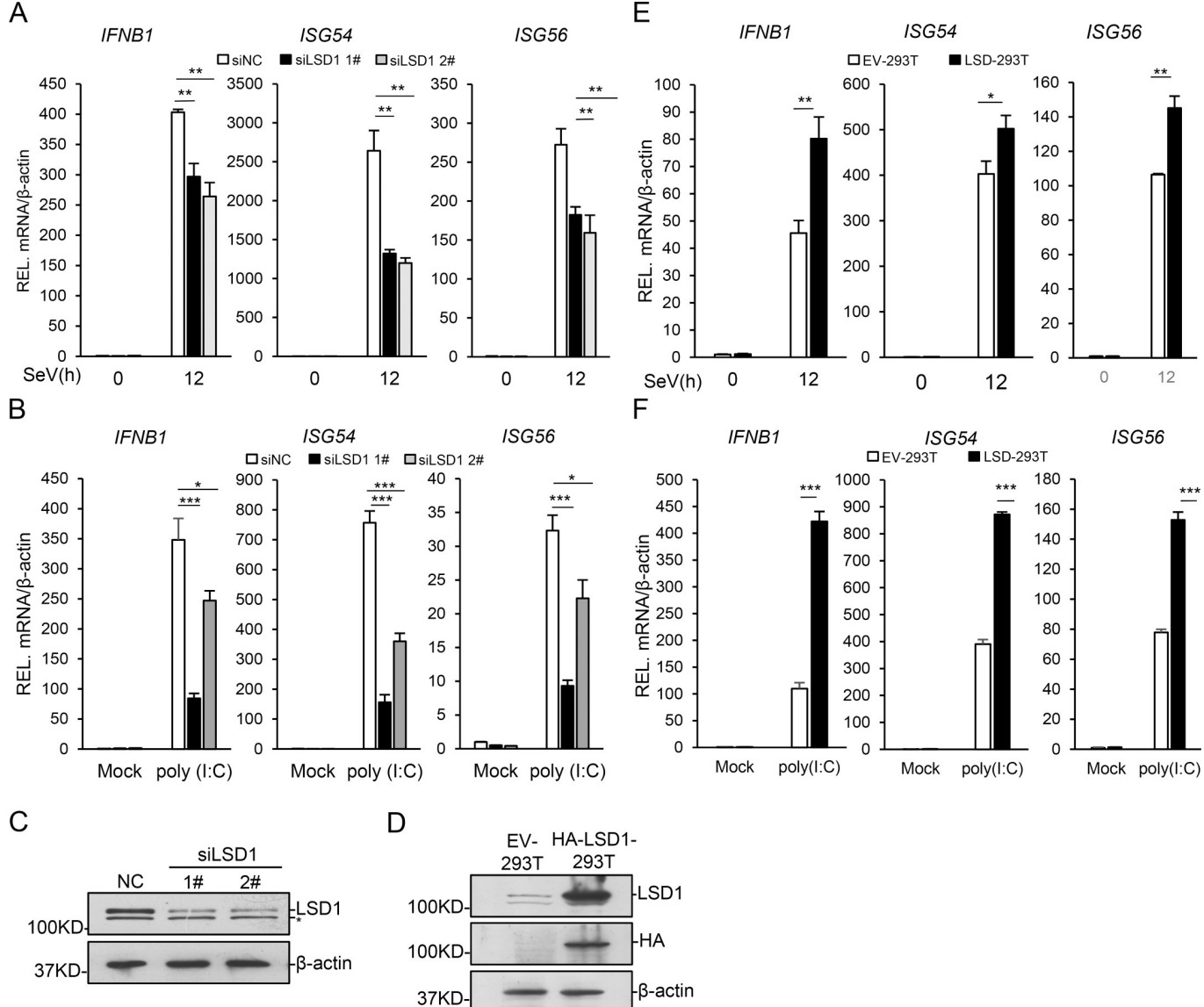

**Fig 1. LSD1 promotes RNA virus-triggered signal pathway. (A)** HEK293T cells were transfected with negative control siRNA (siNC) or LSD1 siRNAs (1#, 2#). The cells were infected with SeV for 12 hr. The relative mRNA levels of *IFNB1*, *ISG54*, and *ISG56* were detected by RT-qPCR. **(B)** HEK293T cells transfected with negative control siRNA (siNC) or LSD1 siRNAs (1#, 2#) and were transfected with poly(I:C) for 10h. The mRNA levels of *IFNB1*, *ISG54*, and *ISG56* were detected by RT-qPCR. **(C)** HEK293T cells were transfected with negative control siRNA (siNC) or *LSD1* siRNAs (1#, 2#), and analyzed by western blotting. **(D)** Stable expression of HA-tagged LSD1 in HEK293T cells was examined by western blotting. **(E)** LSD1-293T and control cells were infected with SeV for 12 hr. The relative mRNA levels of *IFNB1*, *ISG54*, and *ISG56* were detected by RT-qPCR. **(F)** EV-293T and LSD1-293T and control cells were transfected with poly(I:C) for 10h. The relative mRNA levels of *IFNB1*, *ISG54*, and *ISG56* were detected by RT-qPCR. Data are presented as means ± SD of three independent experiments. Student's *t* test was used for statistical calculation. ns, no significance. *$P < 0.05$, **$P < 0.01$, and ***$P < 0.001$.

These data together demonstrated that LSD1 is involved in regulation of *IFNB1* and *ISGs* expression.

## LSD1 impairs virus replication in cultured cells

To investigate whether LSD1 affects virus replication in cells, we used an engineered VSV virus with GFP integration in the genome. HEK293T cells were transfected with *LSD1* siRNAs and infected with VSV-GFP. The depletion of *LSD1* significantly increased the amount of green fluorescence, which represented VSV-GFP in cells (Fig 2A). Western blotting was performed to

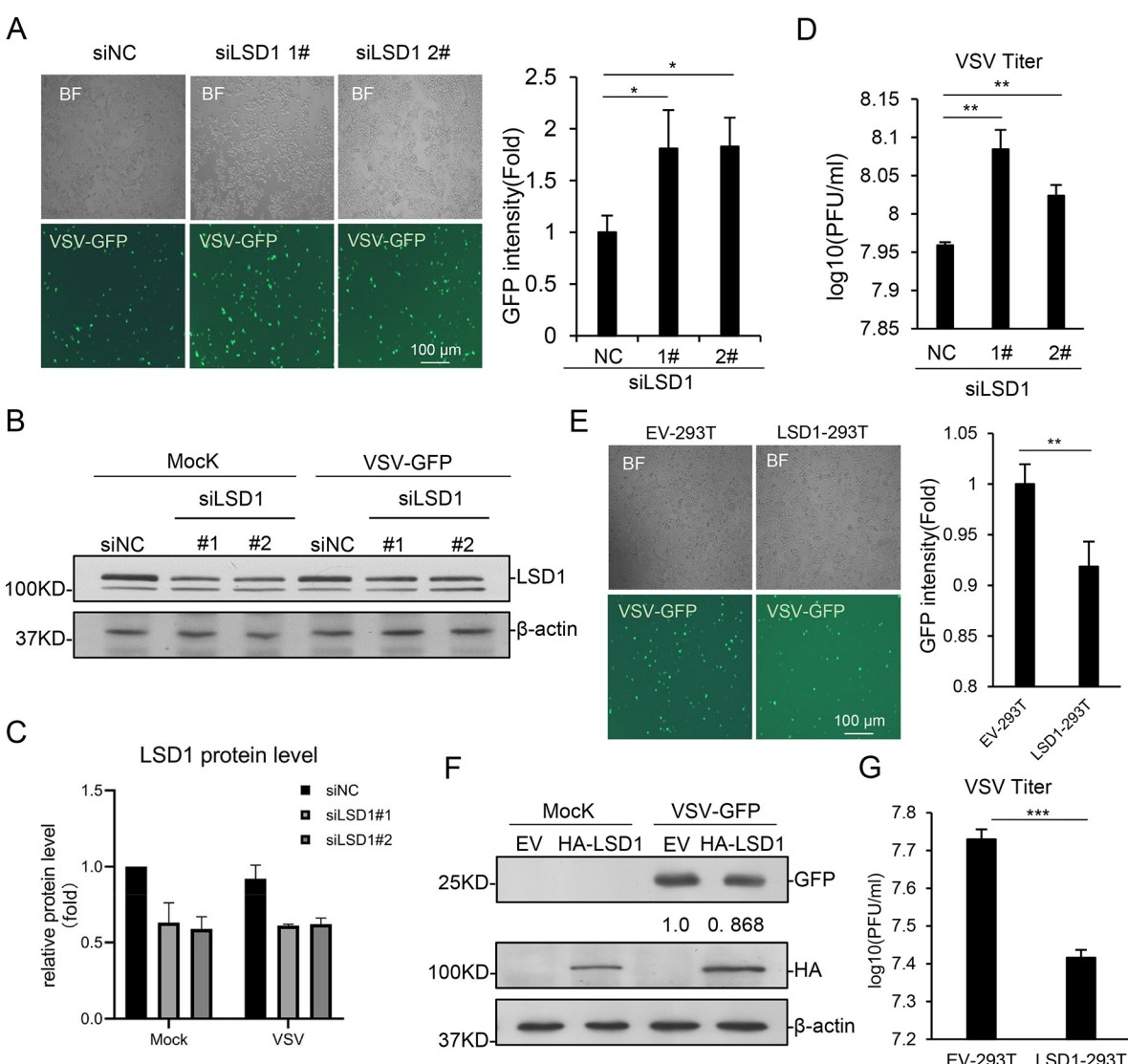

**Fig 2. LSD1 inhibits virus replication. (A-D)** HEK293T cells transfected with siNC or siLSD1 (1#, 2#) were infected with VSV-GFP (MOI = 1) for 8 hr. **(A)** The replication of VSV-GFP were observed by fluorescence microscopy (left). The intensity of GFP fluorescence were analyzed by ImageJ (right). **(B&C)** Infected cells were lysed and analyzed by western blotting with indicated Abs. The relative amount of LSD1 was quantified with ImageJ software. **(D)** VSV titers in supernatants of HEK293T cells were tested by plaque forming unit (PFU) assay. **(E-G)** LSD1-293T and control cells were infected with VSV-GFP (MOI = 1) for 8h. **(E)** The replication of VSV-GFP were observed by fluorescence microscopy (left). The intensity of GFP fluorescence were analyzed by ImageJ (right). **(F)** Infected cells were lysed and analyzed by western blotting with indicated Abs. **(G)** VSV titers in supernatants of the HEK293T cells were tested by plaque forming unit (PFU) assay. Data are presented as means ± SD of three independent experiments. Student's t test was used for statistical calculation. ns, no significance. *P < 0.05, **P < 0.01, and ***P < 0.001.

verify GFP expression (Fig 2B and 2C). The viral titers of VSV were measured with plaque assay which also showed that the virus amount was higher in LSD1-depleted cells (Fig 2D). Consistently, exogenous expression of LSD1 significantly decreased the amount of VSV-GFP fluorescence and GFP protein (Fig 2E and 2F), as well as the viral titers (Fig 2G). Experiments in A549 cells showed that LSD1 also regulates IFNB1 expression in A549 cells (S1A–S1C Fig). These data indicated that LSD1 represses the replication of RNA viruses in cells.

## LSD1 interacts with RIG-I

To figure out the mechanisms of LSD1 in regulating RIG-I signal pathway, we first analyzed the activation of TBK1 and IRF3, two critical factors of RIG-I pathway, by measuring their phosphorylation with western blot. LSD1 depletion decreased the poly(I:C)-induced TBK1 and IRF3 phosphorylation in HEK293T cells (Figs 3A and S1D). Consistently, the poly(I:C)-induced TBK1 and IRF3 phosphorylation were higher when LSD1 was over expressed (Figs 3B and S1E). These suggest LSD1 is involved in TBK1 and IRF3 phosphorylation upon pathway activation and perhaps at the upstream of TBK1.

Then an immunoprecipitation survey was carried out to identify the proteins associated with LSD1 in RIG-I pathway. The results indicated that LSD1 specifically interacts with RIG-I (Fig 3C). The interaction between endogenous LSD1 and RIG-I was successfully detected with or without SeV treatment (Fig 3D). The interaction between exogenous expressed LSD1 and VISA was also detected, but not that between endogenous proteins. So, we think probably LSD1 only interacts with RIG-I, but not with VISA at the endogenous level. To characterize the domains responsible for the interaction, we made a series of RIG-I truncations (Fig 3E). The data of immunoprecipitation assays indicated that LSD1 interacted both CARD and dCARD fragments and full-length RIG-I, among which CARD fragment was the strongest (Fig 3F). Four LSD1 truncations were then constructed, and the co-immunoprecipitation results suggested LSD1 interacts with RIG-I independent of its SWIRM domain (Fig 3G and 3H). Collectively, these data indicate that LSD1 interacts with multiple domains of RIG-I.

## LSD1 enhances K63-linked polyubiquitination of RIG-I

RIG-I is a key molecule to initiate the cascade of RIG-I signaling pathway, and its K63-linked polyubiquitination is required for MAVS/VISA recruitment and downstream events. Modifications on RIG-I by different forms of polyubiquitination chains emerged as critical for the pathway activation [15,56–60]. We investigated the impact of LSD1 on RIG-I polyubiquitination by different ubiquitin chains. HA-LSD1, Flag-RIG-I, and Myc-ubiquitin (Ub) were co-expressed in cells, and *in vivo* ubiquitination assay was performed with anti-Flag IP and anti-Myc western blotting. The result indicated that exogenous expressed LSD1 promotes RIG-I polyubiquitination dramatically (Fig 4A). Further characterization indicated that LSD1 promotes the K63-linked polyubiquitination chain on RIG-I, but not the K48-linked (Fig 4B). To investigate whether LSD1 regulates MDA5, we used EMCV, one virus known to activate signaling through MDA5, to treat cells and *LSD1* knockdown did not cause difference in *IFNB1* expression (S2A Fig). Further studies showed that although over-expressed LSD1 interacted with MDA5, it could not enhance MDA5 ubiquitination (S2B–S2D Fig). These indicated that LSD1 specifically functions on RIG-I, but not MDA5. A gradual increasing dose of HA-LSD1 was expressed in cells, and the result indicated the level of RIG-I K63-linked polyubiquitination increased along with LSD1 expression (Fig 4C). We then investigated whether LSD1 regulates the interaction between RIG-I and VISA interaction. HA-LSD1, HA-VISA and Flag-RIG-I plasmids were co-transfected in cells and co-immunoprecipitation was performed. The result showed that LSD1 promoted RIG-I and VISA interaction (Fig 4D).

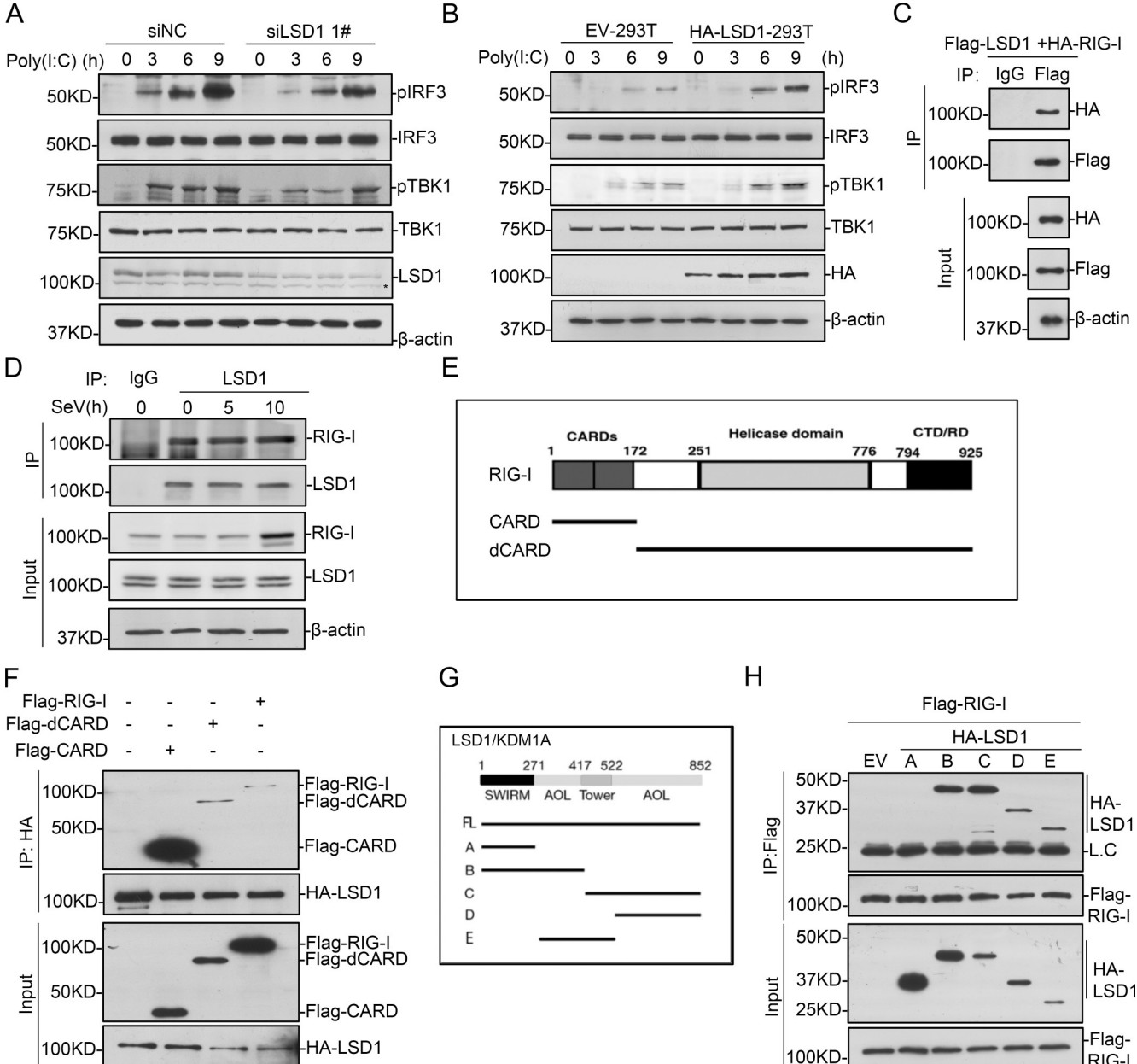

**Fig 3. LSD1 is associated with RIG-I. (A)** NC and LSD1 knock-down HEK293T cells were transfected with poly(I:C) for indicated hours. The phosphorylation levels of IRF3 and TBK1 were analyzed by western blotting. **(B)** LSD1-293T and control cells were transfected with poly(I:C) for indicated hours, and the phosphorylation levels of IRF3 and TBK1 were analyzed by western blotting. The relative amount of phosphorylated IRF3 and TBK1 was quantified with ImageJ software and labelled under the corresponding bands. **(C)** HEK293T cells were transfected with Flag-tagged LSD1 and HA-tagged RIG-I plasmid for 24 hr. Lysates were co-immunoprecipitated with anti-Flag body and immunoblotting analysis with indicated Abs. **(D)** HEK293T cells were infected with SeV for the indicated time and co-immunoprecipitated and immunoblotting were performed with indicated Abs. **(E)** Schematic of full-length RIG-I and its truncation. **(F)** HEK293T cells were transfected with Flag-tagged RIG-I full length or truncations and HA-tagged LSD1 for 24 hr, followed by co-immunoprecipitation and immunoblotting analysis with indicated Abs. **(G)** Schematic of full-length LSD1 and its truncated mutants. **(H)** HEK293T cells were transfected with HA-tagged LSD1 or its truncations and Flag-tagged RIG-I for 24 hr, followed by co-immunoprecipitation and immunoblotting analysis with indicated Abs.

To map the residues of RIG-I whose polyubiquitination are regulated by LSD1, we first determined the domains regulated by LSD1. Two truncations of RIG-I, Flag-CARD or Flag-dCARD, were co-expressed with HA-LSD1 and Myc-Ub in cells, respectively. The result of

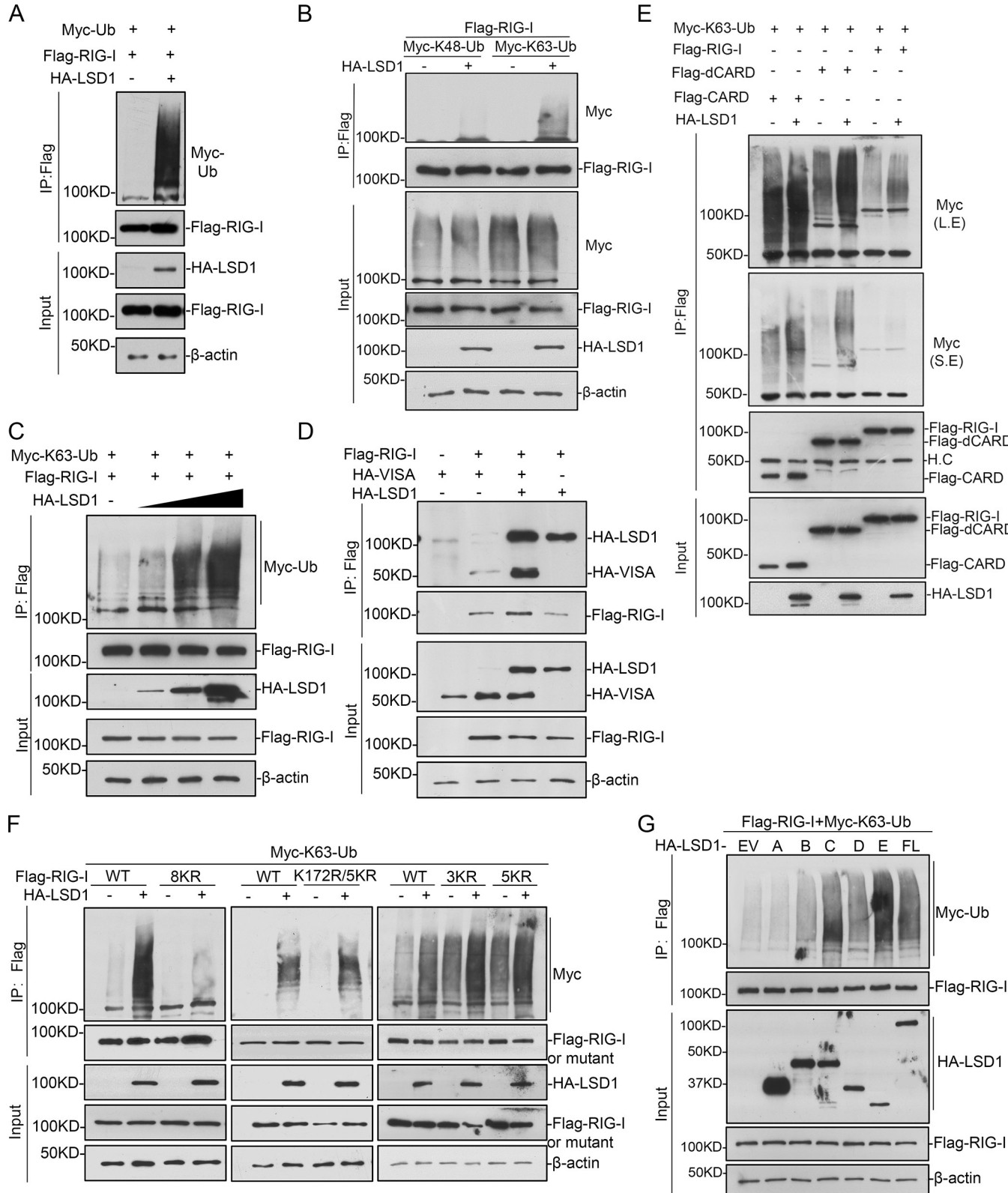

**Fig 4. LSD1 enhances RIG-I K63-linked ubiquitination.** **(A)** Flag-RIG-I and Myc-Ub together with control or HA-LSD1 were expressed in HEK293T for 24 hr, followed by immunoprecipitation and immunoblotting analysis as indicated. **(B)** The experiments were performed as in (A), except K48 only or K63 only ubiquitin plasmid was used instead of wild-type Ub. **(C)** HEK293T cells were transfected with Flag-RIG-I and Myc-K63-Ub together with a gradient dose of

HA-LSD1 for 24 hr, followed by immunoprecipitation and immunoblotting analysis with the indicated antibodies. **(D)** HEK293T were transfected with indicated plasmids for 24 hr, followed by immunoprecipitation and immunoblotting analysis. **(E)** HEK293T cells were transfected with Myc-K63O-Ub (ubiquitin with all lysine mutated to arginine except K63) and Flag-RIG-I or its truncations together with a control or HA-LSD1 expression plasmid for 24 hr, followed by immunoprecipitation and immunoblotting analysis as indicated. The relative amount of ubiquitinated RIG-I was quantified with ImageJ software and labelled under the corresponding bands. L.E. means long exposure, and S.E. for short exposure. **(F)** HEK293T cells were transfected with Myc-K63O-Ub and Flag-RIG-I or its mutants together with control or HA-LSD1 for 24 hr, followed by immunoprecipitation and immunoblotting as indicated. 5KR refers to K849R, K851R, K888R, K907R and K909R; 3KR for K154R, K164R and K172R; 8KR for all the above 8 sites. The relative amount of ubiquitinated RIG-I was quantified with ImageJ software and labelled under the corresponding bands. **(G)** HEK293T cells were transfected with Flag-RIG-I and Myc-K63O-Ub together with control or HA-LSD1 truncations for 24 hr, followed by immunoprecipitation and immunoblotting analysis as indicated.

ubiquitination assay suggested exogenous expressed LSD1 promotes polyubiquitination of both truncations (Figs 4E and S3A), consistent with the previous immunoprecipitation result (Fig 3F). According to previous reports [25,30–33,61], 8 lysine-specific point mutants were constructed. Lysine at 154, 164 and 172 are reported to be targeted for CARD polyubiquitination, and lysine at K849, K851, K888, K907 and K909 are for dCARD polyubiquitination. However, all the 8 single-point mutants (K to R) only showed very tiny decrease of polyubiquitination promoted by LSD1. Then 3KR (K154R, K164R and K172R), 5KR (K849R, K851R, K888R, K907R and K909R) and all 8KR (all 8 lysines to arginines) mutants were constructed. Only the 8KR mutant showed significant impairment of LSD1-enhanced RIG-I polyubiquitination (Figs 4F and S3B–S3D). To map the domains of LSD1 responsible for RIG-I polyubiquitination, HA-LSD1 truncations, Flag-RIG-I and Myc-Ub were co-expressed in HEK293T cells for in vivo ubiquitination assay. All the truncations improved RIG-I polyubiquitination except SWIRM domain (Fig 4G). We further examined the functions of LSD1 truncations, and found that LSD1-C and -E fragments could up-regulate *IFNB1* expression and repress VSV, but not LSD1-A, which was consistent with their ability in regulating RIG-I ubiquitination (S4A–S4C Fig). Taken together, our results proved that LSD1 promotes the activation of RIG-I signaling pathway through facilitating RIG-I K63-linked polyubiquitination and VISA/MAVS recruitment. These data suggest that exogenous expressed LSD1 is capable of promoted ubiquitination of multiple residues, and binding and ubiquitination of RIG-I by LSD1 seem to be two separate events.

## RIG-I polyubiquitination regulated by LSD1 is not dependent on demethylation

LSD1 is reported to demethylate mono-/di-methylation of histone H3K4 and H3K9, as well as non-histone proteins [40,62]. To determine whether the demethylase activity is required for LSD1 in regulating RIG-I polyubiquitination, an enzymatic dead K661A mutant of LSD1 is constructed [63,64]. Co-immunoprecipitation of HA-LSD1 and Flag-RIG-I showed no difference between LSD1 wild type (WT) and mutant (Fig 5A). Ubiquitination assay indicated K661A was able to increase RIG-I polyubiquitination, similar to LSD1 WT (Fig 5B). Overexpression of WT or K661A LSD1 in HEK293T cells both enhanced SeV- or poly(I:C)-induced transcription of *IFNB1*, *ISG54* and *ISG56* (Figs 5C, S5A and S5B), and no significant difference was detected between WT and K661A mutant. For further validation, LSD1 and K661A mutant were expressed in *LSD1* knockdown cells, and quantitative RT-PCR results indicated both WT and mutant LSD1 rescued the phenotype caused by *LSD1* knockdown (S5C Fig). Both WT and mutated LSD1 suppressed viral replication as well (S5D Fig). However, K661A mutant showed weaker inhibitory effect than WT (S5D Fig). As recently reported, LSD1 restricts RNA virus infection by erasing IFITM3-K88 monomethylation, which is an IFN-induced antiviral protein disrupting the entry of viruses to host cells [40,50]. These data indicated LSD1 represses virus replication through multiple pathways, and LSD1 functions on RIG-I polyubiquitination independent of its demethylase activity.

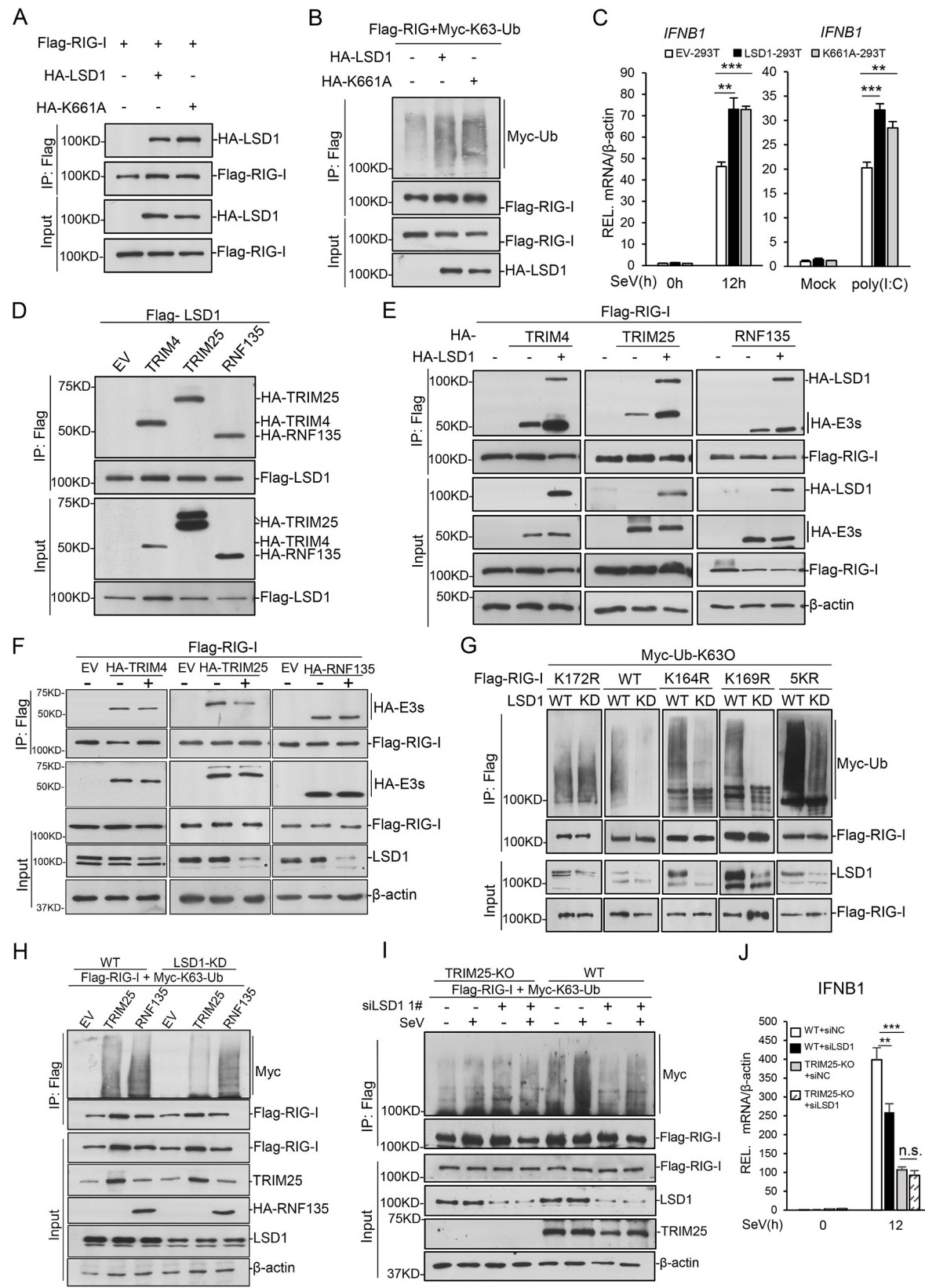

**Fig 5. LSD1 enhances the interaction between RIG-I and E3 ligases. (A)** Flag-RIG-I plasmids and HA-LSD1 or K661A mutant were expressed in HEK293T for 24h. Lysates were co-immunoprecipitated and immunoblotted analysis with indicated Abs. **(B)** HEK293T cells were transfected with Flag-RIG-I and Myc-K63O-Ub together with a control or HA-LSD1 or K661A mutant expression plasmid for 24 h, followed by immunoprecipitation and immunoblotting analysis with the indicated Abs. **(C)** Control 293T, LSD1-293T and K661A-293T cells were infected with SeV for 12 hr (left) or transfected with poly(I:C) for 10h (right). The relative mRNA levels of *IFNB1* were detected by RT-qPCR. **(D)** HEK293T cells were transfected with Flag-tagged LSD1 and three HA-tagged E3 ligases respectively for 24 hr, followed by co-immunoprecipitation and immunoblotting analysis with indicated Abs. **(E)** HEK293T cells were transfected with the indicated plasmids for 24 hr, followed by co-immunoprecipitation and immunoblotting analysis. **(F)** HEK293T cells were lipo-transfected with siNC or *LSD1* siRNA. After 12 hr, HEK293T cells were transfected with indicated plasmids for 24 hr, and interactions between RIG-I and E3 ligases were examined by co-immunoprecipitation. The relative amount of LSD1 was quantified with ImageJ software and labelled under the corresponding bands. * means nonspecific bands. **(G)** Myc-Ub-K63O (ubiquitin with all lysine mutated to arginine except K63) and Flag wild type and indicated mutants were co-expressed in HEK293 wild type or *LSD1* knockdown cells. Ubiquitination assay was then performed. 5KR refers to K849R, K851R, K888R, K907R and K909R. **(H)** Wide-type and *LSD1*-KD HEK293T cells were transfected with Flag-RIG-I and Myc-K63O-Ub together with control plasmid or HA-TRIM25, HA-RNF135 for 24 hr, followed by immunoprecipitation and immunoblotting as indicated. **(I&J)** Wide-type and *TRIM25* KO cells were transfected with negative control siRNA (siNC) or *LSD1* siRNAs 1#. Cells were transfected with indicated plasmids for 24h, and infected with SeV for 10 hr before immunoprecipitation and immunoblotting (I); or cells were treated with SeV for 12 hr, *IFNB1* relative mRNA level of were detected by RT-qPCR (J). Data are presented as means ± SD of three independent experiments. Student's *t* test was used for statistical calculation. ns, no significance. $^{*}P < 0.05$, $^{**}P < 0.01$, and $^{***}P < 0.001$.

## LSD1 promotes the interaction between RIG-I and specific E3 ligases

E3 ubiquitin ligases TRIM4, TRIM25 and RNF135 are reported involved in RIG-I K63-linked polyubiquitination under RNA virus invasion [25,30–33,61]. TRIM4 targets K154, K164 and K172 of RIG-I, K172 for TRIM25, and K849, K851, K888, K907 and K909 for RNF135. Based on these discoveries, we speculated that LSD1 might regulate the interaction between RIG-I and ubiquitin K63-linked E3 ligases. Flag-LSD1 and HA-E3 ligases were co-expressed in cells respectively, and co-immunoprecipitation assay indicated exogenous expressed LSD1 interacts with all the three E3 ligases (Fig 5D). Co-immunoprecipitation assays suggested all of LSD1 truncations interacted with three E3 ligases except its SWIRM domain (S6A Fig). The co-immunoprecipitation of LSD1 and E3 truncations showed LSD1 probably interacted with E3 ligases through their C-terminals but not RING domain (S6B and S6C Fig). Then, we studied the interaction between RIG-I and TRIM4, TRIM25 or RNF135 with or without LSD1 exogenous expression. The results showed that when co-expressed with LSD1, RIG-I pulled down more TRIM4, TRIM25 and RNF135 (Fig 5E). It is consistent with the result about polyubiquitinated residue survey (Fig 4F). When exogenous expressed, LSD1 showed a pan effect of RIG-I polyubiquitination enhancement, as a result all the 8 lysine polyubiquitination could be promoted by LSD1. Moreover, all the three E3 ligases did not affect the interaction of RIG-I and LSD1 (S6D Fig), which indicates LSD1 interacts with RIG-I without E3 ligases.

It is quite surprising that LSD1 regulates the interaction between RIG-I and all three E3s. Thus, we investigated the interaction of RIG-I with three E3s in absence of endogenous LSD1. The result indicated when LSD1 was knocked down, RIG-I recruited much less TRIM25, whereas TRIM4 and RNF135 were not affected (Figs 5F and S7A), suggesting that at the endogenous protein level, LSD1 mainly promotes the interaction between RIG-I and TRIM25. TRIM25 mainly catalyzes poly-ubiquitination on lysine 172 on RIG-I [25]. To confirm the above result, we generated LSD1 knockdown HEK293 cell line with CRISPR technique, and examined the poly-ubiquitination level in the absence of LSD1. The results showed that LSD1 deficiency impaired poly-ubiquitination of RIG-I wild type and most of the tested mutants, but not K172R, which further supported that LSD1 mainly regulates LSD1 through TRIM25 and K172 ubiquitination in vivo (Figs 5G and S7B). We then utilized a TRIM25 knockout cell line and found that RIG-I poly ubiquitination promoted by LSD1 was impaired in the absence of TRIM25 (S7C Fig); and LSD1 expression could enhance RIG-I polyubiquitination catalyzed by low amount of TRIM25 (S7D Fig). We then expressed TRIM25 or RNF135 in wild type or

LSD1 knockdown cells, and found that the ability of TRIM25 to ubiquitinate RIG-I was greatly impaired without LSD1, but RNF135 did not change obviously (Fig 5H). We further knocked down *LSD1* in *TRIM25* knockout cells, and found that *LSD1* deficiency decreased RIG-I poly-ubiquitination and *IFNB1* expression in wild type cells, but not in *TRIM25* knockout cells (Fig 5I and 5J). Co-immunoprecipitation assays showed that interaction between LSD1 and RIG-I was not dependent on TRIM25 or RIG-I ubiquitination (S8A–S8C Fig). These data together demonstrated LSD1 promoted RIG-I K63-linked polyubiquitination through enhancing the interaction of RIG-I and ubiquitin E3 ligase TRIM25, probably not TRIM4 or RNF135.

## Lsd1 deficiency inhibits RNA virus-triggered RIG-I signaling in vivo

We next investigated the role of Lsd1 in antiviral response *in vivo*. We generated Lsd1 conditional knockout mice by crossing *Lsd1*$^{f/+}$ with lyz2-cre mice. We isolated lung fibroblasts (MLFs), bone marrow-derived dendritic cells (BMDCs) and bone marrow-derived macrophages (BMDMs) from the above mouse model and infected with VSV and SeV, respectively. Quantitative RT-PCR experiments indicated that *Lsd1* deficiency impaired virus-induced transcription of *Ifnb1*, *Isg56*, *Il-6* and *Cxcl10* genes in MLFs, BMDMs and BMDCs (Figs 6A and 6B; S9A–S9D). Consistently, *Lsd1* deficiency inhibited the phosphorylation of Tbk1 and Irf3 induced by SeV infection in MLFs, BMDCs and BMDMs (Figs 6C, S9E and S9F). Collectively, these data suggest that LSD1 is required for the proper activation of RNA virus-stimulated RIG-I signaling pathway in primary murine cells.

To further explore the roles of LSD1 in host defense against viral infection in vivo, we used VSV to infect *Lsd1*$^{f/+}$ with and without Lyz2-cre mice by tail-intravenous injection. Quantitative RT-PCR results indicated VSV-induced transcription of *Ifnb1*, *Isg54*, *Il-6* and *Cxcl10* in blood were impaired in *Lsd1*$^{f/+}$*Lyz2-cre* mice compared to the control mice. Similar results were obtained in mice liver, lung and spleen (Figs 6D, S10A–S10C). Moreover, viral titer and number of genomic copies of VSV in the lung and spleen were much higher in *Lsd1*$^{f/+}$*Lyz2-cre* mice than those in the control group (Fig 6E and 6F). Meanwhile, pathological analysis revealed that VSV infection resulted in increased alveolar wall thickening, and severe edema in the lungs of *Lsd1*$^{f/+}$*Lys2-cre* mice compared with *Lsd1*$^{f/+}$ mice (Fig 6G). *Lsd1* deficient mice also had much lower survival rate upon VSV challenge (Fig 6H). Together, these results suggest that LSD1 is required for RIG-I signaling and host defense against RNA virus *in vivo*.

## Discussion

In the current study, we reveal a novel role of LSD1 in regulating antiviral RIG-I signaling. LSD1 promotes the activation of RIG-I pathway and represses the replication of RNA virus *in vivo* and *in vitro*. It interacts with RIG-I and enhances K63-linked polyubiquitination of RIG-I to promote signal transduction. Shan *et al.* reported LSD1 restricts RNA virus invasion by erasing IFITM3-K88 monomethylation [40]. Our study demonstrates that LSD1 not only regulates one of the antiviral products, but also acts as a critical positive regulator of RIG-I signaling pathway, which actually affects the expression of much more anti-viral genes. Thus, our study illustrates it as one of the important players in innate immunity.

Many literatures have shown that LSD1 exerts its roles through its demethylase activity. Our study proved that LSD1 promotes RIG-I polyubiquitination through facilitating the interaction between RIG-I and TRIM25. Both wild type and mutant LSD1 were able to inhibit virus replication compared to control samples. Based on our results, we speculate it is possible that the interaction between LSD1 and RIG-I changes RIG-I conformation and then facilitates the recognition by ubiquitin E3 ligases. Recently, several studies have also showed that LSD1

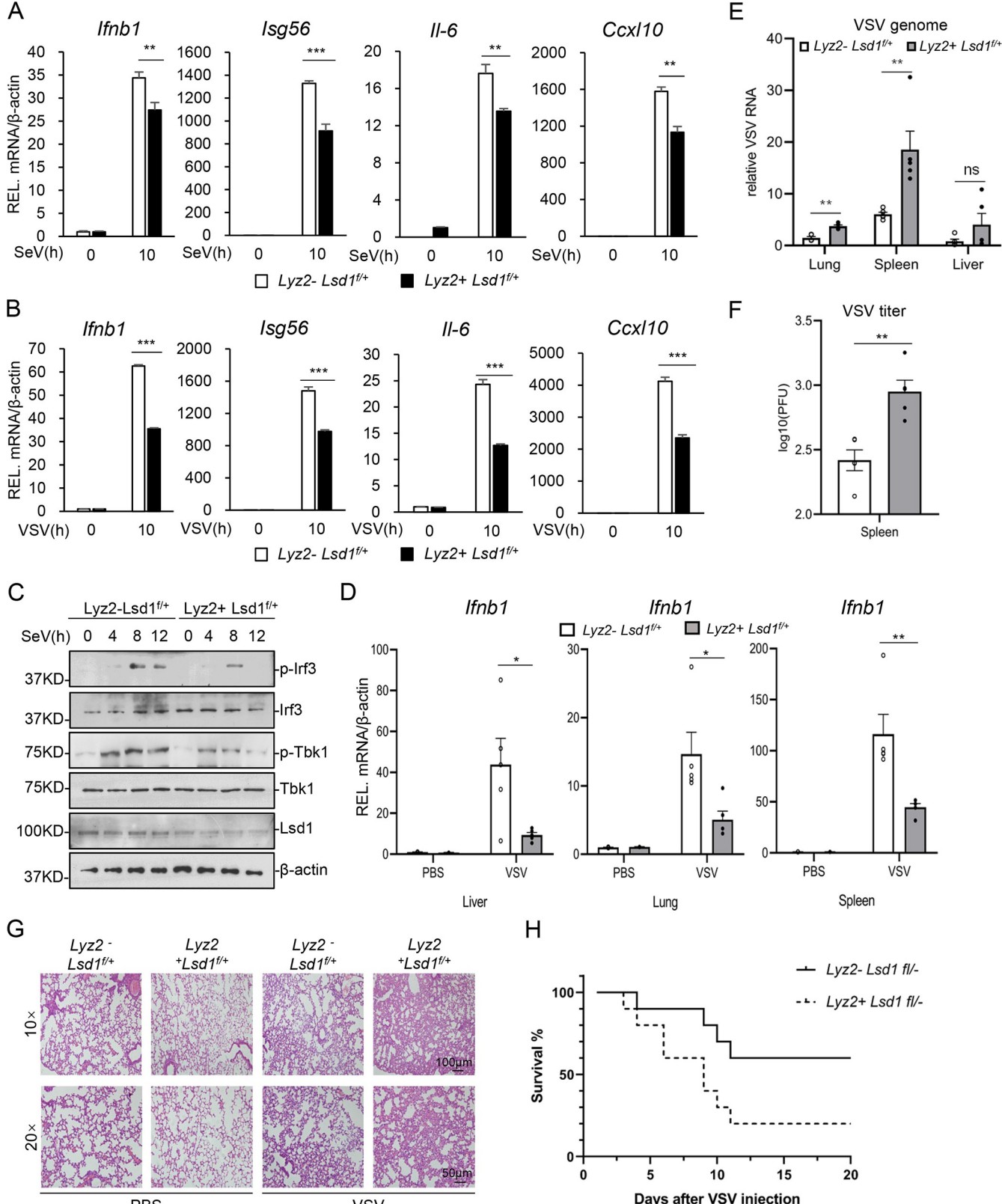

**Fig 6. Lsd1 is required for Rig-i-mediated innate immune response. (A&B)** *Lyz2- Lsd1*<sup>fl/+</sup> and *Lyz2+ Lsd1*<sup>fl/+</sup> MLFs were left uninfected or infected with SeV(A) or VSV(B) for 10h. The relative mRNA levels of *Ifnb1*, *Isg56*, *Il-6*, *Cxcl10* were detected by RT-qPCR. **(C)** *Lyz2- Lsd1*<sup>fl/+</sup> and *Lyz2+ Lsd1*<sup>fl/+</sup> MLFs were

left un-infected or infected with SeV for the indicated times, followed by immunoblotting analysis with indicated Abs. **(D)** 8-week-old *Lyz2- Lsd1*$^{fl/+}$ and *Lyz2 + Lsd1*$^{fl/+}$ mice were tail-intravenous injected with VSV at 10^8 PFU per mouse (n = 5 for each genotype group) for 12h. Total RNA from mice liver, lung and spleen were extracted and the relative mRNA level of *Ifnb1* were detected by RT-qPCR. **(E&F)** VSV genome RNA (E) and VSV titers(F) of tissues from infected mice in (A-C) were tested with qPCR and PFU analysis, respectively. **(G)** Hematoxylin and eosin staining of lung sections from mice in (A). Scale bars, 100 µm (for 10×) and 50 µm (for 20×). **(H)** *Lsd1*$^{fl/+}$ *Lyz2-cre-* and *Lsd1*$^{fl/+}$ *Lyz2-cre+* mice were intraperitoneal injected with VSV at 10^8 PFU per mouse (n = 10 for each genotype group). The survival rates of mice were recorded. Data are means ± SEM and are representative of three biological replicates. Student's *t* test was used for statistical calculation. ns, no significance. $^{*}P < 0.05$, $^{**}P < 0.01$, and $^{***}P < 0.001$.

functions independent of its enzyme activity [65]. These studies together expand the functions of LSD1 beyond a demethylase.

LSD1 was generally considered as a protein functioning in nuclei; however, our immune-staining assays showed that it was also localized both in nuclei and cytoplasm in some cells. Then it is possible for it to interact with RIG-I, which is localized in cytoplasm. Then it is interesting what the regulatory mechanism is to control LSD1 localization, which is critical for its cellular functions.

Our data showed exogenous expression of LSD1 enhanced TRIM4, TRIM25 and RNF135-linked RIG-I polyubiquitination. However, LSD1 depletion mainly impaired the interaction between RIG-I and TRIM25, but not TRIM4 or RNF135. The results observed with exogenous expressed LSD1 were probably some artifacts caused by gene over expression. Interestingly, we found that LSD1 interacts with RIG-I without virus treatment, and it promoted RIG-I poly-ubiquitination but did not activate *IFNB1* expression without virus. These indicated that RIG-I polyubiquitination activated by LSD1 alone is not sufficient to activate the pathway, and some other factors are required, which should be investigated in the future studies. Taken together, our results indicate that LSD1 regulates RIG-I signaling mainly *via* recruiting TRIM25 and promoting K172-linked polyubiquitination of RIG-I. Of course, we cannot exclude the possibility that LSD1 may function through other E3 ligases during certain circumstance.

To sum up, our study demonstrates LSD1 as a positive regulator of RIG-I signaling through promoting RIG-I K63 polyubiquitination and the interaction between RIG-I and TRIM25. Considering the abnormal activation of immune response is connected with many diseases, such as COVID-19, our study may provide important theoretical information for developing novel clinical strategies.

## Supporting information

**S1 Fig. LSD1 regulates IFNB1 expression in A549 cells.**
(PDF)

**S2 Fig. LSD1 is not involved in IFNB1 expression activated by EMCV.**
(PDF)

**S3 Fig. Functional studies of LSD1 truncations.**
(PDF)

**S4 Fig. Functions of LSD1 truncations.**
(PDF)

**S5 Fig. LSD1's function is independent of demethylase activity.**
(PDF)

**S6 Fig. LSD1 interacts with ubiquitin E3 ligases of RIG-I.**
(PDF)

**S7 Fig. LSD1 mediates TRIM25-dependent RIG-I polyubiquitination.**
(PDF)

**S8 Fig. LSD1 interacts with RIG-I independent of TRIM25.**
(PDF)

**S9 Fig. Effects of Lsd1-deficiency on RIG-I signaling in murine primary cells.**
(PDF)

**S10 Fig. Lsd1 Is Required for Rig-i-Mediated Innate Immune Response in mice.**
(PDF)

**S1 Table. List of primers for qPCR.**
(DOCX)

## Acknowledgments

We appreciate Dr. Hong-Bing Shu of Wuhan University for sharing reagents and project discussion.

## Author Contributions

**Conceptualization:** Lin-Gao Ju, Yuan Zhu, Min Wu, Lian-Yun Li.

**Funding acquisition:** Min Wu, Lian-Yun Li.

**Investigation:** Qi-Xin Hu, Hui-Yi Wang, Lu Jiang, Zhen Wang.

**Methodology:** Qi-Xin Hu, Hui-Yi Wang, Bo Zhong.

**Project administration:** Min Wu, Lian-Yun Li.

**Resources:** Bo Zhong.

**Software:** Chen-Yu Wang.

**Supervision:** Min Wu, Lian-Yun Li.

**Validation:** Qi-Xin Hu, Zhen Wang.

**Writing – original draft:** Zhen Wang.

**Writing – review & editing:** Min Wu, Lian-Yun Li.

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
