## [Decision Letter · Decision Letter 0]

18 May 2021

Dear Dr. Wu,

Thank you very much for submitting your manuscript "Histone demethylase LSD1 promotes RIG-I poly-ubiquitination and anti-viral gene expression" for consideration at PLOS Pathogens. As with all papers reviewed by the journal, your manuscript was reviewed by members of the editorial board and by several independent reviewers. In light of the reviews (below this email), we would like to invite the resubmission of a significantly-revised version that takes into account the reviewers' comments.

We cannot make any decision about publication until we have seen the revised manuscript and your response to the reviewers' comments. Your revised manuscript is also likely to be sent to reviewers for further evaluation.

Sincerely,

Stacy M Horner

Associate Editor

PLOS Pathogens

Jing-hsiung James Ou

Section Editor

PLOS Pathogens

Kasturi Haldar

Editor-in-Chief

PLOS Pathogens

orcid.org/0000-0001-5065-158X

Michael Malim

Editor-in-Chief

PLOS Pathogens

orcid.org/0000-0002-7699-2064

Reviewer's Responses to Questions

**Part I - Summary**

Reviewer #1: The manuscript by Hu et al. reports the identification of histone demethylase LSD1 as a positive regulator of the RIG-I signaling pathway. Silencing and ectopic expression of LSD1 confirmed its role in regulating antiviral ISG/cytokine responses and VSV replication. Biochemical evidence showed that LSD1 interacted with RIG-I and promoted its K63-linked polyubiquitination, and this effect was independent of its demethylase activity and was likely due to promoting interaction between RIG-I and E3 ligases, particularly TRIM25. Lastly, the role of LSD1 in the regulation of cytokine responses and viral replication in vivo was corroborated by using VSV challenge of myeloid cell lineage conditional KO mice.

The manuscript extends the antiviral function of LSD1 to RLR-mediated innate immune responses and presents a good amount of data exploring the potential mechanism by which LSD1 regulates RIG-I. The study is expected to be of high interest to the field. However, some of the mechanistic insights are obscured by an abundance of negative data that were likely due to overexpression artifacts and are difficult to interpret. Therefore, a defined role for LSD1 in regulating RIG-I-mediated antiviral innate immunity should be further strengthened by additional experiments.

Reviewer #2: RIG-I is one of the central innate immune sensors of invading RNA (and some DNA) viruses. Tri (or di) phosphorylated RNA produced upon viral infection is recognized by RIG-I and a signaling cascade ultimately leading to the induction of type-I IFNs (via IRF3/NF-kB) and other pro-inflammatory cytokines (NF-kB) is initiated. For full activation of RIG-I it has to be post-translationally modified by K63-linked ubiquitination, which is mediated by e.g. TRIM25, RNF135, TRIM4 and/or MEX3C.

In this manuscript Hu and co-authors propose that the methyltransferase LSD1 positively regulates RIG-I signaling by enhancing the interaction of RIG-I with the E3 ligase TRIM25, which promotes K63-linked polyubiquitination and the interaction with MAVS/VISA. Accordingly, LSD1 overexpression increases SeV/poly(I:C)-induced expression of antiviral genes and inhibits replication of VSV. Knockdown experiments show the opposite effect, which was also confirmed in LSD1-deficient mice. Further, the function of LSD1 is not dependent on its methylase activity.

In summary, in an updated version this manuscript would provide novel and interesting insight into the regulation of the innate immunity sensor RIG-I. The study is well planned, defines a phenotype and novel (albeit moderate) regulator of RLR signaling -that is relevant in vivo- but the mechanistic analyses fall short and do not (yet) support the main conclusions.

Reviewer #3: The authors of this manuscript found that the histone demethylase LSD1 could interact with RIG-I, and that knockdown of LSD1 expression reduced poly(I:C) or SeV-induced IFN-beta and ISG production. In addition, overexpression of LSD1 enhanced the K63-ubiquitination of RIG-I. The authors also showed that LSD1 could be involved in the interaction between RIG-I and TRIM25. Finally, the research applied conditional LSD1 knockout mice to demonstrate the role of LSD1 in IFN-beta and antiviral response. The findings of this research are quite interesting in the research field of innate immune response. Although the authors provided many biochemical evidences to correlate LSD1 and RIG-I activation in the manuscript, several points should be further clarified before drawing the conclusion.

**Part II – Major Issues: Key Experiments Required for Acceptance**

Reviewer #1: 1. The silencing effect of LSD siRNA was marginal in Fig 2B (and, in fact, in many other panels), though there seemed to be a statistically significant phenotype on VSV replication (Fig 2C). Is this related to the cell type (293T) used in which endogenous LSD1 was hard to deplete? Have the authors tried other cell types such as lung epithelial cells or fibroblasts that are also more physiologically relevant to VSV infection? It appears that the entire study, except for the ex vivo data in Fig 6, was performed in HEK 293T cells. The data on the phenotype on cytokine induction and viral replication should be confirmed using at least one more relevant cell type, such as A549 cells, or primary cells preferably.

2. In Fig 4, the potentiating effect of LSD1 on RIG-I polyubiquitination was only examined in the context of LSD1 overexpression, which should be validated by LSD1 silencing. Specifically, since LSD1 interacted strongly with the 2CARD (Fig 3F), which is known to undergo robust K63-linked polyubiquitination without the requirement of Ub overexpression, the silencing experiment should be preferentially conducted using the 2CARD construct. For mapping ubiquitination sites in the context of LSD1 overexpression (Fig 4F), the experimental setting should include key E3 ligase co-expression (or stimulation with a RIG-I ligand), as there was essentially not even a difference between the WT and nKR mutants in ubiquitination levels in the absence of LSD1. In addition to the nKR controls, the focus should be placed on K172R and K788R as both sites are the major ubiquitination sites by TRIM25 and Riplet, respectively (Gack et al. 2007; Oshiumi et al. 2013). A limiting concentration of LSD1 might also have to be experimentally determined so that it does not overwhelm the system thereby masking any effect on the KR mutants (see also point #5).

3. The data in Fig 5G are somewhat misleading as WT RIG-I was the least ubiquitinated compared to other KR mutants; samples should be loaded on the same gel for an unbiased comparison. Nevertheless, Fig 5G does clearly point to TRIM25 as the LSD1 target and the K172 as the major site of ubiquitination promoted by LSD1. However, how do these data reconcile with that in Fig 4F then? As the authors were proposing an interesting model where LSD1 potentially bridges the interaction between TRIM25 and RIG-I, additional data showing that LSD1 can sensitize a limiting amount of TRIM25 to catalyze RIG-I ubiquitination to a similar level as does a high concentration of TRIM25 only would further strengthen their proposed model.

Reviewer #2: In general, the effects of LSD1 are very moderate. The authors could consider using LSD1 KO cells, if feasible for their experiments.

Fig. 3: LSD1 was described as mainly, if not exclusively nuclear. How is an association with the cytoplasmic RIG-I/TRIM25 complex rationalized? The authors need to show that in a cell LSD1/RIG-I/TRIM25 can be found in a similar localization. This issue has to be addressed experimentally.

Fig. 4A: Ubiquitination of RIG-I is dramatically increased by LSD1 expression; however, it seems to not activate RIG-I as in cells overexpression of LSD1 does not stimulate ISG expression. Compare also Fig. 5C. Can the authors explain why this is the case, and whether the ubiquitination of RIG-I they observe then even contributes to RIG-I activation?

Fig 4E/F and text lines 255-256: Text and Fig 4E and 4F are not consistent. 4E: It does not look like LSD1 promotes Flag-CARD ubiquitination, the levels seem to be equal with and without LSD1. In contrast 4F indicates that LSD1 ubiquitinates CARD (3KR) and dCARD (5KR) to comparable levels. How do the authors explain that the classically by TRIM25 ubiquitinated lysines can be removed and still enhanced ubiquitination is observed for LSD1 overexpression if its mechanism is TRIM25 dependent? Fig. 4E: How can ubiquitination of RIG-IdCARD be promoted if RIG-IdCARD does not interact with LSD1? The passage in the manuscript (lines 255/256) states that 4E is consistent with the IP from Fig 3F, which is not the case. The current inconsistencies suggest to me that the proposed mechanism is not completely correct. The authors need to provide more evidence to support their conclusions and explain why some of the current data is not supporting this hypothesis. For example: In TRIM25KO cells, does LSD1 still promote ubiquitination of RIG-I? Is the RIG-I/TRIM25/LSD1 complex formed in the absence of RIG-I ubiquitination (e.g. using the 8R mutant).

Fig. 4: For all ubiquitination assays please show the input myc (ubiquitin) blots. This is mandatory to exclude differences in the input levels that could affect the IP.

Reviewer #3: 1. The authors showed that the expression of IFN-beta and ISGs was changed by knockdown or overexpression of LSD1 in the Figure 1. However, it cannot conclude that “LSD1 plays an important role in promoting RIG-I signaling” (line 194) from the results. From this part of the results, it can only conclude that LSD1 is involved in regulation of IFN-beta and ISG expression.

2. From the results in Figure 3, the authors could only conclude that LSD1 could affect TBK1 and IRF3 phosphorylation, not really “functions upstream of TBK1” (lines 214-215), unless the authors determine the LSD1’s role in IFN-beta expression under TBK1 overexpression.

3. The results showed that the endogenous LSD1 interacts with RIG-I even at “0 h” after SeV infection (Fig. 3D, lane 2). Does this mean that endogenous LSD1 could interact with RIG-I and facilitate RIG-I K63 ubiquitination/activation without any stimulation? However, overexpression of LSD1 itself seems not to activate IFN-beta (Fig. 1F). The authors have to explain the controversial results.

4. The results for determining the region of LSD1 for RIG-I interaction showed that multiple regions in LSD1 can interact with the RIG-I (Fig. 3H). Does overexpression of each LSD1 construct (B-E) enhance IFN-beta/ISG expression under poly(I:C) or SeV stimulation and suppress viral replication?

5. The construct B of the truncated LSD1 interacts with RIG-I well (Fig. 3H). However, the truncated LSD1 does not enhance much K63 ubiquitination (Fig. 4G). The authors should explain this phenomenon.

6. The conclusion described in lines 229-230 should be modified, unless the authors can map the site in LSD1 for RIG-I interaction and show that the RIG-I binding site mutant of LSD1 loses the ability for stimulating RIG-I polyubiquitination.

7. In the Fig. 4E, the overexpression of LSD1 did not enhance polyubiquitiation of RIG-I-CARDs (Fig 4E, lane 1 vs 2). It has been known that the ubiquitination of K172 in RIG-I CARDs is mainly mediated by TRIM25. The result showed that LSD1 could interact with TRIM25 (Fig. 5D and E). Why LSD1 overexpression did not enhance polyubiquitiation of RIG-I-CARDs?

8. In Supp Fig. 1C, the knockdown efficiency of LSD1 looks perfect (no LSD1 was detected in lanes 3 and 4), not like in the Fig. 1C, 2B and 3A. However, the reduction of SeV-activated IFN-beta/ISG expression looks similar. The authors may have to perform a dose-dependent experiment (knockdown or overexpression of LSD1) to conclude that LSD1 is involved in regulation of IFN activation.

9. In the Fig. 5F, the knockdown efficiency was around 50%. However, the TRIM25-RIG-I interaction was almost abolished (lane 5 vs 6). Could the authors explain it?

10. The authors might show the localization of LSD1 and RIG-I in cells upon SeV infection to demonstrate these proteins can be colocalized in the physiological condition.

**Part III – Minor Issues: Editorial and Data Presentation Modifications**

Reviewer #1: 1. It is quite clear from the data that LSD1, by silencing or overexpression, affected IFN/ISG expression (Fig 1). However, whether this effect was through specific regulation of RIG-I was insufficiently proven. Silencing LSD1 upon RIG-I or MDA5 overexpression should be performed to determine specificity. Relating to this point, in Fig 3C, MDA5 or MAVS should be included as controls to confirm RIG-I specificity.

2. From the mapping data in Fig 3F, the conclusion in lines 225-227 stating LSD1 interacted weakly with ΔCARD seems incorrect. Full-length RIG-I, despite being expressed more than ΔCARD, bound even lesser to LSD1. How to explain this?

3. The ex vivo and in vivo data in Fig 6 convincingly show the antiviral effect of LSD1. It would be nice if the authors could perform additional data to link these data to RIG-I/TRIM25, for example by performing 1 experiment in cells from these mice. Moreover, why was the tail-intravenous route chosen over intranasal for VSV challenge? Were there any data on animal survival?

Reviewer #2: Fig. 2B/5F: the LSD1 KD efficacy is relatively low and varies a lot, please quantify for 3 independent replicates.

Fig. 2B Please remove the panel or at least add a VSV protein, not just GFP.

Fig. 2 It would have been more convincing if a second virus would have been included.

Fig3 A and B: Please quantify the levels of p-IRF3/IRF3 and p-TBK1/TBK1 of 3 independent experiments

Fig. 3C: How can the authors say specifically, if only RIG-I was tested, was a larger panel checked?

Fig 3F: Indicate that HA-LSD1 was overexpressed. Further, the LSD1 Input is not equal.

How can you explain that LSD1 interacts with Flag-CARD, but not the full-length RIG-I? Please at least address and discuss.

Fig 4E/F: Please quantify the ubiquitination levels of 3 independent experiments.

Fig 4B: Please use another replicate where the actin level of the (-) K63 lane are not drastically lower

Fig 5F: It seems like HA-RNF135 decreases LSD1 expression (there is nearly no LSD1 detectable in the Input). Was this effect observed in all 3 experiments or was it just observed on this occasion?

Fig. 5F: Bubble in blot at the crucial band, has to be replaced in a final version.

Fig. 6: Show survival of mice when challenged.

Abstract line 25: RIG-I stands for retinoic acid-inducible gene I and not for retinoic acid-inducible I gene receptor

Line 332: Lsd1 - LSD1

Line 340: Tbk1, Irf3 - TBK1, IRF3

Line 651: rig.i, Lsd1 - RIG-I, LSD1

Please improve the consistency of the manuscript. Sometimes the IPs are labeled as IP: Flag (e.g. Fig 4D) and sometimes as Flag IP (e.g Fig 4F), qPCRs are labelled with either IFNB1.. (e.g Fig. 1) or Ifnb1…- (e.g Fig 6, Fig S3), WBs with either p-IRF3 (e.g. Fig 3) or p-Irf3 (e.g. Fig S3).

Please remove the overstatements in line 196, line 176

Reviewer #3: 1. The materials and methods of the manuscript were not described clearly:

(1) The Sendai virus and GFP-tagged VSV were used in the research, but there is no description in the materials and methods. How much SeV was used for stimulation? How was the GFP-VSV generated or obtained?

(2) The source or reference for the siRNA library was not mentioned in the text.

(3) The reference for calcium phosphate transfection is suggested to be added.

(4) What kind of poly(I:C) was used in the research?

2. In the Fig. 5D, two bands were detected for TRIM25, as marked by the authors. Could the authors explain it?

3. The authors should denote what the Myc-Ub-K63O is in the legends of Fig. 4E and Fig. 5G.

PLOS authors have the option to publish the peer review history of their article (what does this mean?). If published, this will include your full peer review and any attached files.

Reviewer #1: No

Reviewer #2: No

Reviewer #3: No
---

## [Editor Report · Decision Letter 1]

19 Aug 2021

Dear Dr. Wu,

Thank you very much for submitting your manuscript "Histone demethylase LSD1 promotes RIG-I poly-ubiquitination and anti-viral gene expression" for consideration at PLOS Pathogens. As with all papers reviewed by the journal, your manuscript was reviewed by members of the editorial board and by several independent reviewers. The reviewers appreciated the attention to an important topic. Based on your response to the first round of reviews, we are likely to accept this manuscript for publication, providing that you modify the manuscript according to our recommendations, indicated here.

(1) Much of the data provided in the response to reviewers is quite compelling, and I would suggest that it should be shown in the manuscript or supplemental data as it increases the rigor, significance, and impact. Specifically, I would suggest including Response Figure 1, 2B, 3, 4, 5, 7, 8, 10.

(2) In addition, please add a discussion of how the cytoplasmic RIG-I and the often nuclear LSD1 could be interacting to the discussion. I don’t think that you need to show the data of Response Figure 6, just describe what might be happening or that you acknowledge that its perplexing and/or interesting.

(3) A discussion of how binding and ubiquitination of RIG-I by LSD1 are two separate events should be included, either in the results (lines 290) or as a point in the discussion, as this important point could be confusing to some.

(4) Thank you for including the quantifications in Fig. 2B, 3A, 3B, 4E, 4F, 5F. Please show the SD of the quantifications of all three replicates to give the reader an idea of the variation among replicates (for example, 0.63-/+ .2).

[1] A letter containing a detailed list of your responses to the editorial comments, and a description of the changes you have made in the manuscript.

Sincerely,

Stacy M Horner

Associate Editor

PLOS Pathogens

Jing-hsiung James Ou

Section Editor

PLOS Pathogens

Kasturi Haldar

Editor-in-Chief

PLOS Pathogens

orcid.org/0000-0001-5065-158X

Michael Malim

Editor-in-Chief

PLOS Pathogens

orcid.org/0000-0002-7699-2064

Reviewer Comments (if any, and for reference):

Figure Files:

Data Requirements:

Reproducibility:

References:

---

## [Editor Report · Decision Letter 2]

26 Aug 2021

Dear Dr. Wu,

We are pleased to inform you that your manuscript 'Histone demethylase LSD1 promotes RIG-I poly-ubiquitination and anti-viral gene expression' has been provisionally accepted for publication in PLOS Pathogens.

Best regards,

Stacy M Horner

Associate Editor

PLOS Pathogens

Jing-hsiung James Ou

Section Editor

PLOS Pathogens

Kasturi Haldar

Editor-in-Chief

PLOS Pathogens

orcid.org/0000-0001-5065-158X

Michael Malim

Editor-in-Chief

PLOS Pathogens

orcid.org/0000-0002-7699-2064
---

## [Editor Report · Acceptance letter]

1 Sep 2021

Dear Dr. Wu,

We are delighted to inform you that your manuscript, "Histone demethylase LSD1 promotes RIG-I poly-ubiquitination and anti-viral gene expression," has been formally accepted for publication in PLOS Pathogens.

Best regards,

Kasturi Haldar

Editor-in-Chief

PLOS Pathogens

orcid.org/0000-0001-5065-158X

Michael Malim

Editor-in-Chief

PLOS Pathogens

orcid.org/0000-0002-7699-2064